# HIV, hepatitis B virus, and hepatitis C virus co-infection among HIV positives in antiretroviral treatment program in selected hospitals in Addis Ababa: A retrospective cross-sectional study

Eleni Seyoum[1,2]*, Meaza Demissie[3], Alemayehu Worku[4], Andargachew Mulu[5], Alemseged Abdissa[5], Yemane Berhane[2]

1 Institute of Public Health, University of Gondar, Gondar, Ethiopia, 2 Epidemiology and Evaluation Department, Addis Continental Institute of Public Health, Addis Ababa, Ethiopia, 3 Public Health Department, Addis Continental Institute of Public Health, Addis Ababa, Ethiopia, 4 Department of Preventive Medicine, School of Public Health, College of Health Sciences, Addis Ababa University, Addis Ababa, Ethiopia, 5 Bacterial and Viral Diseases Research Directorate, Armauer Hansen Research Institute, Addis Ababa, Ethiopia

* Eleniseyoum45@gmail.com

**Data Availability Statement:** All relevant data are within the manuscript and its Supporting Information files.

## Abstract

### Introduction

HIV co-infection with hepatitis B (HIV-HBV) and hepatitis C (HIV-HCV) is known to affect treatment outcomes of antiretroviral therapy (ART); however, its magnitude is not well documented. We aimed to determine the magnitude of HIV-HBV and HIV-HCV co-infections simultaneously in people living with HIV (PLHIV) enrolled in ART care in Addis Ababa.

### Methods

We reviewed the medical records of adults $\geq$15 years who were receiving ART care in three high burden hospitals in Addis Ababa. Baseline clinical and laboratory test results were extracted from medical records. Co-infection was determined based on hepatitis B surface antigen (HBsAg) and hepatitis C virus antibody (anti-HCV) tests obtained from the medical records. A multivariable logistic regression model was used to identify the risk factors for hepatitis B and C co-infections.

### Results

A total of 873 HIV-positive participants were included in this study. The median age of the participants was 37.5 years, and 55.7% were women. Overall, HIV-HBV co-infection was 5.96% (95% CI: 4.56–7.74), and HIV-HCV co-infection was 1.72% (95% CI: 1.03–2.83). The multivariable logistic regression showed that the male sex was the most independent predictor for viral hepatitis B co-infection with an odds ratio of 2.42(95% CI:1.27–4.63).

**Funding:** The author(s) received no specific funding for this work.

**Competing interests:** All authors have declared that no competing interests exist.

**Abbreviations:** ART, Antiretroviral Treatment; HBV, Hepatitis B Virus; HCV, Hepatitis C Virus; PLHIV, People Living with HIV; HBsAg, Hepatitis B surface antigen; WHO, World Health Organization; ARV, Antiretroviral Therapy; HMIS, Health Information System; ALT, Alanine Aminotransferase concentration; AST, Aspartate Aminotransferase; IQR, Inter-quartile Range; CI, Confidence Interval.

However, HIV-HCV co-infection did not show a significant association in any of the sociode-mographic data of the participants.

## Conclusion

HIV co-infection with hepatitis B was moderately high in individuals enrolled in ART care in Addis Ababa. Men had significantly higher HIV-HBV co-infection. HIV co-infection with hepatitis C was relatively low. Strengthening integrated viral hepatitis services with HIV care and treatment services should be emphasized to improve patient care in health facilities.

## Introduction

HIV-associated morbidity and mortality have declined in resource-limited countries owing to the rapid scale-up of antiretroviral therapy [1]. However, co-infection with hepatitis B virus (HBV) and hepatitis C virus (HCV) has emerged as a clinical and public health challenge [2]. Studies have shown an increase in the number of liver-related deaths among antiretroviral users [2–4].The guidelines of the World Health Organization (WHO) in 2016 recommended the early detection and screening of viral hepatitis in people living with HIV (PLHIV) at the initiation of antiretroviral therapy (ART) [5]. Globally, less than 5% of PLHIV know their HBV or HCV status [6].Although Africa is known to harbor most of the HIV, HBV, and HCV infected people, little is known about viral hepatitis co-infection status in HIV-positives enrolled in ART care.

In Ethiopia, more than 669,000 people were living with HIV; 11,500 people died from an AIDS-related illness, and 71% of PLHIV were treated by the end of 2019 [7]. Viral hepatitis is endemic, with an estimated national hepatitis B prevalence of 9.4% in 2017 [8]. There is no recent national population-based HCV study; however, population-based studies from the northwest and southwest of the country showed hepatitis C prevalence of 1% in 2017 [9] and 1.9% in 2018 [10].

As in most parts of the world, there is a general insufficiency in the diagnosis of HBV and HCV as in Ethiopia, which remains a challenge in explaining the magnitude and pattern of co-infections and providing the necessary treatment to HIV positives. Studies in some hospitals in Addis Ababa have estimated co-infections [11, 12]. Understanding the magnitude of HIV-HBV and HIV-HCV co-infection is necessary to prioritize care and treatment services for co-infected individuals.

The lack of continuous studies to estimate HBV and HCV in Ethiopia remains a serious challenge in determining the pattern of co-infection. Therefore, we aimed to determine the magnitude of HIV-HBV and HIV-HCV co-infection in PLHIV who were on ART in Addis Ababa.

## Methods

### Study design and setting

A retrospective cross- sectional medical record review was carried out in three big hospitals in Addis Ababa; Zewditu Hospital, Alert Hospital, and Black Lion Hospital. The three hospitals together serve 16,400 ART clients. Addis Ababa, the capital city of Ethiopia, has a population of more than five million people [13]. Addis Ababa is the second highest (3.5%) HIV prevalent

city, with an estimated 125,000 PLHIV in 2020. In addition, one-fourth of the total number of PLHIV in the country receive ART in health facilities in Addis Ababa [7].

The antiretroviral treatment program in Ethiopia is provided based on the national guidelines, which were adopted from the WHO guidelines issued in 2016. The guideline recommends screening all HIV-positives for both hepatitis B and hepatitis C viruses at ART initiation [14, 15].

However, the guidelines are not fully practiced in most health facilities that provide ART services in the country. The recommended testing is a serological test for HB surface antigen (HBsAg) and anti-HCV antibodies. The guideline also recommends a first-line drug combination regimen of antiviral agents (tenofovir and entecavir) that are active against HBV. Direct acting drugs that treats HCV are very expensive and not easily accessible, even for those who can afford to pay for it until recently. In response to this, the Federal Ministry of Health has promoted and included HCV direct-acting drugs in the essential drug list. Following that private healthcare providers have started procuring direct-acting drugs to treat HCV. The government is also working to expand affordable access to direct-acting drugs through increasing procurement, private partnerships, and awareness. The second national five years (2021–2025) Viral Hepatitis Strategic Plan also included subsidized procurement of direct-acting drugs as one strategic direction to improve the health of the nation.

The outcome of this study was the magnitude of HIV-HBV co-infection and HIV-HCV co-infection. Adults $\geq$ 15 years who had documented viral hepatitis B and viral hepatitis C test results enrolled in ART care from September 2012 to December 2018 (started routine viral hepatitis screening) were eligible. Study participants without viral hepatitis test results were excluded from the analysis. A comparison was made between those who had test results and those without test results to check for selection biases using selected demographic and clinical characteristics of age, sex, education, and marital status.

## Sample size determination and sampling procedures

The sample size for the study was determined using a single proportion formula, which required 256 and 240 HBV and HCV positives, respectively. The following assumptions were included in the calculation:Co-infection prevalence of HIV-HBV and HIV-HCV was 0.052 and 0.055, respectively [16], margin of error, d = 0.03, Z = 1.96, and a 95% CI. However, we included all eligible adults $\geq$ 15 years with documented viral hepatitis test results enrolled in ART care from September 2012 to December 2018 to increase the precision of the estimate and be able to conduct internal comparisons. A total of 873 (Zewditu hospitals, n = 523, ALERT, n = 186, and Black Lion hospital, n = 164) HIV positives with documented viral hepatitis B and viral hepatitis C were included in this study.

## Data extraction procedures

The data extraction form was developed in English and pretested in two health facilities that were not part of the study sites. Four data abstractors (two BSc nurses and two data managers) were trained in the selected study hospitals. The data collectors were team-up in two groups with a data manager and a nurse in one group to abstract clinical and laboratory data from the patient folder directly into a tablet. An experienced supervisor with a master's degree in Public Health joined the data abstractor team.

The folders of the HIV positives was the source of information for this study. The client folder included a follow-up card in which the HIV positive ' visits were recorded at ART initiation and during the follow-up visits. Laboratory results and history intake forms were attached to the patient folder. The patient folder is kept in the Health Information System (HMIS)

archive, which is a dedicated card room where all patient cards are kept independent of the patient's HIV status. The ART register and the electronic database were used to obtain a list of the unique ART numbers and eligible adults for the study.

The ART data manager sorted 20–30 eligible unique ART numbers from the database and sent them to a runner (HMIS card person) to remove the patient folder from the HMIS for data abstraction during the study period.

The data extracted from the medical records of HIV-positive included sociodemographic characteristics (age, sex, marital status, education, and religion), baseline laboratory and clinical characteristics, hepatitis B surface antigen (HBsAg) test, and anti-hepatitis C virus (anti-HCV) test. Extracted data were uploaded daily to the REDCAP server after the supervisor checked for completeness of the data. REDCAP is a secure web application for building and managing online surveys and databases. The data were collected from November 12, 2019, to March 15, 2020. Data were exported from REDCAP to STATA14 for further cleaning, data management, and analysis.

## Statistical analysis

The background characteristics of the study participants were described with the median and interquartile range (IQR) for numeric variables and frequency and proportion for categorical variables. The normality of the distribution was tested first to determine the type of statistical test. For numeric variables, the Kruskal-Wallis equality-of-populations rank test or Wilcoxon test was used as appropriate. For categorical outcome variables, the chi-square test was used. Statistical significance was set at $p < 0.05$. The co-infection of HIV-HBV and HIV-HCV was calculated using proportions and presented with a 95% confidence interval. Multivariable logistic regression was used to assess risk factors for HIV-HBV and HIV-HCV, and the odds ratio (OR) with 95% CI was presented.

## Ethics approval and consent to participate

This study was approved by the Institutional Review Board of the University of Gondar in Ethiopia and the Armauer Hansen Research Institute, Addis Ababa, Ethiopia. No consent was required from participants. No external persons accessed the patient's folder, and no patient identifier was included in our data set. The patient folder that was taken from the HMIS room for data abstraction was kept in a locked file cabinet until it was returned to the HMIS room to maintain participant's confidentiality. The analysis used routinely collected patient data for program monitoring and evaluation purpose, and had no direct patient contact. As this was a retrospective study, the University of Gondar Institutional Review Board waived the need for informed consent. We confirm that all methods have been carried out in accordance with relevant guidelines and regulations.

## Results

### Characteristics of the study participants at ART enrollment

Of the 998 adults eligible for this study, 873 were included in this analysis, and 125 (15%) of the study participants were excluded from the analysis because they did not have documented viral hepatitis test results. Of the 873 participants included, their median age at enrollment was 37.5, with an inter-quartile range (IQR) of–31–45 years, and 55.7% were female. Married participants constituted the highest proportion 408(48.69%) of the study participants. Regarding educational status, 731(86.82%) of the study participants had attained basic education, 296 (35.15%) attained secondary education, and 173(20.55%) attained tertiary education Table 1.

**Table 1. Socio-demographic characteristics of the participants at ART enrollment (September 2012-December 2018), Addis Ababa, n = 873.**

| Variables | n(%) |
|---|---|
| **Age in (years), (n = 873)** | |
| 15–29 | 166 (19.01) |
| 30–44 | 469 (53.72) |
| 45–59 | 198 (22.68) |
| 60+ | 40 (4.58) |
| Median (IQR) | 37.5 (31–45) |
| **Sex, (n = 869)** | |
| Male | 385(44.30) |
| Female | 484(55.69) |
| **Marital status, (n = 838)** | |
| Never married | 210(25.05) |
| Married | 408(48.68) |
| Separated/Divorced | 140(16,70) |
| Widowed/er | 80(9.54) |
| **Education, (n = 842)** | |
| No education | 111(13.18) |
| Primary | 262(31.11) |
| Secondary | 296(35.15) |
| Tertiary | 173(20.54) |
| **Religion, (n = 843)** | |
| Muslim | 97(11.50) |
| Orthodox | 646(76.63) |
| Protestant | 89(10.55) |
| Catholic | 4(0.47) |
| Other | 7(0.83) |

IQR:Inter-quartile range; ART: Antiretroviral Therapy.

## Magnitude of HIV-HBV co-infection

The overall HIV-HBV co-infection was 5.96% (95% CI: 4.56–7.74, n = 52) in PLHIV enrolled for ART care. HBV co-infection was highest in men (8.83%;95% CI: 6.37–12.12, n = 34) than in female participants and 3.72% (95% CI: 2.35–5.83, n = 18, p = 0.0001). In terms of age group, HBV co-infection was the highest in the age group of 45–59 years (7.57% [95% CI: 4.60–12.23,]) and lowest in the age group 15–29 years (4.21% [95% CI: 2.01–8.64, p = 0.626]).

Married study participants had the highest HIV-HBV co-infection of 7.59%(95% CI: 5.39–10.62, p = 0.316), followed by separated/divorced (5.71%;95% CI: 2.86–11.10), and the lowest HBV was reported among those who had never been married (3.80%;95% CI: 1.91–7.47) Table 2.

## Magnitude of HIV-HCV co-infection

HIV-HCV co-infection among HIV-positive enrolled for ART care was 1.72% (95% CI: 1.03–2.83, n = 15). HCV co-infection was highest, 7.50% (95% CI: 2.31–21.71, p = 0.014) in the age group of 60+, followed by the age group 15–29 years 2.41% (95% CI: 0.89–6.30). With regard to the marital status of HIV positiveswith HCV, widowed/er showed the highest HCV co-infection, 2.50% (95% CI: 0.61–9.72) followed by never married, 2.38% (95% CI: 0.99–5.63),

**Table 2. HIV-HBV and HIV-HCV co-infection in HIV positives enrolled in ART care by selected background characteristics (September 2012-December 2018), Addis Ababa, N = 873.**

| Variables | Overall | HIV-HBV Co-infection | | HIV-HCV Co-infection | |
|---|---|---|---|---|---|
| | N | N | Estimated Prevalence (%) (95% CI) | N | Estimated Prevalence (%) (95% CI) |
| Total | 873 | 52 | 5.96(4.56–7.74) | 15 | 1.72(1.03–2.83) |
| Age in (years) | | | | | |
| 15–29 | 166 | 7 | 4.21(2.01–8.64) | 4 | 2.41(0.89–6.30) |
| 30–44 | 469 | 28 | 5.97(4.15–8.52) | 4 | 0.85(0.32–2.25)* |
| 45–59 | 198 | 15 | 7.57(4.60–12.23) | 4 | 2.02(0.75–5.30)* |
| 60+ | 40 | 2 | 5.00(1.18–18.87) | 3 | 7.50(2.32–21.71) |
| Sex | | | | | |
| Male | 385 | 34 | 8.83(6.37–12.12)* | 7 | 1.82(0.87–3.77) |
| Female | 484 | 18 | 3.72(2.35–5.83) | 7 | 1.45(0.69–3.01) |
| Marital status | | | | | |
| Never married | 210 | 8 | 3.81(1.91–7.47) | 5 | 2.38(0.99–5.63) |
| Married | 408 | 31 | 7.59(5.39–10.62) | 4 | 0.98(0.37–2.59 |
| Separated/Divorced | 140 | 8 | 5.71(2.86–11.10) | 2 | 1.43(0.35–5.62) |
| Widowed/er | 80 | 4 | 5.00(1.85–12.82) | 2 | 2.50(0.61–9.72) |
| Education | | | | | |
| No education | 111 | 6 | 5.40(2.42–11.64) | 1 | 0.90(0.12–6.29) |
| Primary | 262 | 14 | 5.34(3.18–8.84) | 4 | 1.52(0.57–4.02) |
| Secondary | 296 | 24 | 8.11(5.48–11.83) | 3 | 1.01(0.33–3.11) |
| Tertiary | 173 | 8 | 4.62(2.31–9.03) | 4 | 2.31(0.86–6.05) |
| Religion | | | | | |
| Muslim | 97 | 6 | 6.18(2.76–13.26) | 1 | 1.03(0.13–7.18) |
| Orthodox | 646 | 41 | 6.34(4.70–8.51) | 11 | 1.70(0.94–3.05) |
| Protestant | 89 | 3 | 3.37(1.06–10.13) | 1 | 1.12(0.15–8.81) |
| Catholic | 4 | 1 | (**) | - | - |
| Other | 7 | 1 | (**) | - | - |

HBV: hepatitis B virus; HCV: hepatitis C virus; HIV: human immunodeficiency virus; CI: confidence interval

*p-value < 0.05 is considered as statistically significant level (

** Observation smaller than 25)

and the lowest was recorded among the married study participants (0.98% [0.37–2.59]). Male's sex had a higher 1.82(0.87–3.77) HCV co-infection compared with female 1.45(95% CI: 0.69–3.01, p = 0.665) study participants Table 2.

## Factor associated with co-infection

The multivariant logistic regression showed that the male sex was the most independent predictor for viral hepatitis B co-infection with an OR of 2.42(95% CI: 1.27–4.63) Table 3. However, the multivariate logistic regression analysis of HIV-HCV co-infection did not show a significant association with HCV co-infection in all the socio-demographic data of the participants (age groups, sex, educational status, marital status, and religion) Table 4.

## Discussion

We found that the magnitude of HIV-HBV co-infection was 5.96% and HIV-HCV co-infection was 1.72% in adults ≥15 years in Addis Ababa. The factor that was statistically and

**Table 3. Factors associated with viral hepatitis B co-infections in HIV positives enrolled in ART care (September 2012-December 2018) Addis Ababa, n = 873.**

| Variables | Viral Hepatitis B Co-infection | | | |
|---|---|---|---|---|
| | Bivariate | | Multivariable | |
| | Odds ratio | p-value | Odds ratio | p-value |
| **Sex** | | | | |
| **Male** | 2.50(1.39–4.51) | 0.002 | 2.42(1.27–4.63) | 0.007 |
| **Female** | 1 | | 1 | |
| **Age group** | | | | |
| **15–29** | 0.83(0.16–4.18) | 0.828 | 1.30(0.23–7.13) | 0.75 |
| **30–44** | 1.20(0.27–5.25) | 0.803 | 1.34(0.29–6.13) | 0.69 |
| **45–59** | 1.55(0.34–7.09) | 0.567 | 1.37(0.28–6.51) | 0.69 |
| **60+** | 1 | | 1 | |
| **Marital Status** | | | | |
| **Single** | 1 | | 1 | |
| **Married** | 2.07(0.93–4.60) | 0.072 | 2.13(0.92–4.92) | 0.074 |
| **Divorced/Separated** | 1.53(0.56–4.17) | 0.406 | 1.87(0.65–5.35) | 0.242 |
| **Widowed/er** | 1.32(0.38–4.54) | 0.650 | 1.51(0.40–5.62) | 0.535 |
| **Education** | | | | |
| **No Education** | 1 | | 1 | |
| **Primary** | 0.98(0.36–2.64) | 0.981 | 0.78(0.28–2.16) | 0.640 |
| **Secondary** | 1.54(0.61–3.88) | 0.356 | 1.24(0.47–3.25) | 0.662 |
| **Tertiary** | 0.84(0.28–2.51) | 0.767 | 0.63(0.20–2.02) | 0.441 |
| **Religion** | | | | |
| **Orthodox** | 1 | | 1 | |
| **Muslim** | 0.97(0.40–2.35) | 0.952 | 0.99(0.40–2.47) | 0.997 |
| **Protestant** | 0.51(0.15–1.69) | 0.276 | 0.53(0.15–1.81) | 0.317 |
| **Catholic** | 4.91(0.50–48.33) | 0.172 | 7.01(0.596–82) | 0.121 |
| **Other** | 2.45(0.28–20.91) | 0.410 | 2.2(0.24–20.62) | 0.479 |

strongly associated with HBV co-infection was the male sex, which had a higher likelihood of being co-infected.

We demonstrated findings similar to those of other studies conducted in HIV-positive enrolled for ART care in hospital settings. For instance, our finding of HIV-HBV is similar to those of studies from the northern part of the country (5.6%,5.9%, and 5.7%) [17–19]. Overall, our finding of HIV-HBV is also consistent with the findings of other similar studies in Ethiopia, ranging from 4.7% to 7.7% [20–23].

On the other hand, our finding of HIV-HBV is much lower than the study conducted in the eastern parts of the country (11%) in three regional hospitals of Diredawa, Somali, Harari, and in the southern part of the country at Wolita Sodo Hospital [24, 25]. This difference seems to be related to traditional practices, such as female genital mutilation between regions, which may facilitate transmission. For instance, female genital mutilation is most common in the eastern part of the country, with 99% in Somali, 91% in Afar, 82% in Harari, and 75% in Dire Dawa compared with 54% in Addis Ababa or other parts of the country [26].

The HIV-HBV prevalence in this study was moderately high. More than 90% of HBV transmission occurs during childbirth in Africa and Asia [27]. The vaccination against the HBV was not yet in place in Ethiopia until 2006, when the country introduced pentavalent vaccination, which includes a vaccine against HBV at birth 14 years prior. Currently, adults ≥ 15 years were not part of the vaccination cohort. This may have contributed to the moderately

**Table 4. Factors associated with viral hepatitis C co-infections in HIV positives (September 2012-December 2018) Addis Ababa, n = 873.**

| Variables | Viral Hepatitis Co-infection | | | |
|---|---|---|---|---|
| | Bivariate | | Multivariable | |
| | Odds ratio | p-value | Odds ratio | p-value |
| **Sex** | | | | |
| **Male** | 1.26(0.43–3.62) | 0.666 | 1.27(0.32–4.96) | 0.728 |
| **Female** | 1 | | 1 | |
| **Age group** | | | | |
| **15–29** | 0.30(0.06–1.41) | 0.130 | 0.54(0.04–6.38) | 0.630 |
| **30–44** | 0.10(0.02–0.49) | 0.004 | 0.12(0.01–1.51) | 0.101 |
| **45–59** | 0.25(0.05–1.18) | 0.008 | 0.49(0.04–5.16) | 0.558 |
| **60+** | 1 | | 1 | |
| **Marital Status** | | | | |
| **Never married** | 1 | | 1 | |
| **Married** | 0.40(0.10–1.52) | 0.182 | 0.29(0.06–1.43) | 0.131 |
| **Divorced/Separated** | 0.59(0.11–3.10) | 0.537 | 0.29(0.03–2.92) | 0.298 |
| **Widowed/er** | 1.05(0.19–5.53) | 0.953 | 0.48(0.04–5.24) | 0.554 |
| **Education** | | | | |
| **No Education** | 1 | | 1 | |
| **Primary** | 1.70(0.18–15.43) | 0.635 | 1.32(0.13–12.80) | 0.808 |
| **Secondary** | 1.12(0.11–10.94) | 0.918 | 0.88(0.08–9.10) | 0.916 |
| **Tertiary** | 2.60(0.28–23.60) | 0.395 | 0.90(0.07–11,12) | 0.937 |
| **Religion** | | | | |
| **Orthodox** | 1 | | 1 | |
| **Muslim** | 0.60(0.07–4.71) | 0.628 | 1.02(0.12–8.67) | 0.979 |
| **Protestant** | 0.65(0.08–5.14) | 0.688 | 0.92(0.11–7.72) | 0.940 |
| **Catholic** | - | | - | |
| **Other** | - | | - | |

high HIV-HBV co-infection findings in this study. Therefore, the moderately high HIV-HBV found in this study may remain the same for some time until the new vaccinated cohort joins the adult population.

In this study, we also found that HIV-HBV co-infection was twice as high in men than in women in the same age group. A similar significant difference between the sexes in HIV-HBV co-infection was reported in Ethiopia and other African countries [17, 19, 28]. The reason why men have higher HBV levels than women is beyond the scope of this study. However, previous large cohort and 12 years follow-up studies showed that biological factors are more important than environmental or behavioral factors. This explains that the higher HBV in men indicates that females develop HBV antibodies faster than males, which helps them to clear HBsAg early, so most may not progress to chronic HBV. Second, the female sex hormone, estrogen, may be a protective factor for females and androgen as a risk factor for men contributing to higher HBV [29, 30]. This may call for emphasis to encourage men to target HBV screening without compromising efforts to women.

We found a low (1.72%) HIV-HCV co-infection among HIV-positive enrolled in ART care in three Addis Ababa hospitals. Others also found extremely low HIV-HCV co-infections, as low as 1.0%in Addis Ababa, 1.3% in Debretabor Hospital, and 1.1% in Gondar Hospital [11, 31, 32].

In contrast, studies from Hawassa, Bahirdar, and Adawa hospitals showed a high HIV-HCV co-infection ranging from 5% to 11% [33–35]. The Hawassa study indicated that only 9.1% of the HIV-HCV cases were confirmed by polymerase chain reaction and were viremic. Studies have shown that 88% of HCV persists with chronic HCV, as opposed to what is reported in the Hawassa study [36]. Therefore, a direct comparison with this study in the absence of HCV viremic results from our study may not be reasonable.

The higher HIV-HCV prevalence from Adawa and Bahirdar hospitals may be due to the associated risk factors identified, including hospitalization, blood transfusion, and tooth extraction, which may contribute to high HIV-HCV co-infection in the area. Our finding of HIV-HCV co-infection is consistent with the Sub-Saharan Africa estimates of the general population, but much lower than the estimate in the HIV population [37].

The high HIV-HCV co-infection in sub-Saharan Africa can be explained by the prevalent behavioral factors such as injection of drugs in the HIV population in some parts of Africa. For instance, in Western and Central Africa, injecting drug use is common in the HIV population [38], while in Ethiopia, the HIV prevalence in people with injecting drug users is not different from the general population in urban setting. In 2018, co-infection of HIV-HBV and HIV-HCV among people with injecting drug users in Addis Ababa was 5.1% and 2.9% respectively [39].

The limitations of this study are as follows: We extracted hepatitis B and C test results from those who had documented viral hepatitis test, and we excluded those who did not have documented viral hepatitis test results (15%) of the eligible population, which may introduce selection bias. However, we compared HIV-positive who had documented viral hepatitis test versus those without based on their biological and background characteristics and did not find any difference. Second, we could not include some potential confounding variables and possible predictors related to behavior and route of transmission because they were not routinely collected at ART initiation. The absence of data on the route of transmission and behavior may affect the predictors of co-infection but does not alter the magnitude of co-infections. Test kits variation at health facilities with possible variation in sensitivity and specificity could be another limitation of the study.

## Conclusions

HIV co-infection with hepatitis B was moderately high in individuals enrolled in ART care in Addis Ababa. Men had significantly higher HIV-HBV co-infection. HIV co-infection with hepatitis C was relatively low. Strengthening integrated viral hepatitis services in HIV care and treatment services should be emphasized to improve patient care in health facilities, and a more detailed study is needed to understand the factors associated with viral hepatitis B and C.

## Supporting information

**S1 File. Raw data used for analysis.**
(DTA)

## Acknowledgments

We are grateful to the data abstractors and the supervisor at the three hospitals. We also thank Mr. Samuel Ayele from AHRI for organizing and setting up the data abstraction tools on the tablets and facilitating access to the REDCAP server and Dr. Yared Mekonnen for reviewing and editing the manuscript. Finally, we would like to thank the Addis Continental Institute of Public Health and the University of Gondar for giving the opportunity.

## Author Contributions

**Conceptualization:** Eleni Seyoum, Meaza Demissie, Yemane Berhane.

**Data curation:** Eleni Seyoum, Meaza Demissie, Alemayehu Worku, Andargachew Mulu.

**Formal analysis:** Eleni Seyoum, Alemayehu Worku, Yemane Berhane.

**Funding acquisition:** Eleni Seyoum, Alemseged Abdissa.

**Investigation:** Eleni Seyoum.

**Methodology:** Eleni Seyoum, Alemayehu Worku, Andargachew Mulu, Yemane Berhane.

**Project administration:** Eleni Seyoum.

**Resources:** Eleni Seyoum, Alemseged Abdissa.

**Software:** Eleni Seyoum.

**Supervision:** Eleni Seyoum, Meaza Demissie, Alemayehu Worku, Alemseged Abdissa, Yemane Berhane.

**Validation:** Eleni Seyoum, Meaza Demissie, Alemseged Abdissa, Yemane Berhane.

**Visualization:** Eleni Seyoum.

**Writing – original draft:** Eleni Seyoum.

**Writing – review & editing:** Eleni Seyoum, Meaza Demissie, Alemayehu Worku, Andargachew Mulu, Alemseged Abdissa, Yemane Berhane.

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
