## [Decision Letter · Decision Letter 0]

15 Feb 2022

PONE-D-21-34939HIV, hepatitis B virus, and hepatitis C virus Co-infection patients in Addis Ababa Antiretroviral Treatment Program, 2012-2018: a retrospective cohort studyPLOS ONE

Dear Dr. Seyoum,

Thank you for submitting your manuscript to PLOS ONE. After careful consideration, we feel that it has merit but does not fully meet PLOS ONE’s publication criteria as it currently stands. Therefore, we invite you to submit a revised version of the manuscript that addresses the points raised during the review process.

We look forward to receiving your revised manuscript.

Kind regards,

Desalegn Admassu Ayana, Ph.D

Academic Editor

PLOS ONE

Journal Requirements:

2. Please provide additional details regarding participant consent. Specifically, if the need for consent was waived by the ethics committee, please include this information. Thank you.

We declar that there was no direct fund that the authors received. However,  the data collection fee at the three hospitals was covered thrugh AHRI by Federal HIV/AIDS Prevention and Contorl Office. The source of fund had no role in the design or conduct of the study; the collection, management, analysis, and interpretation of the data; or the preparation, review, or approval of the manuscript. The authors had full access to all the data in the study and takes the responsibility for the integrity of the data and the accuracy of the data analysis. The authors declars that there was no fund received from any organization. 

7. We note that you have included the phrase “data not shown” in your manuscript. Unfortunately, this does not meet our data sharing requirements. PLOS does not permit references to inaccessible data. We require that authors provide all relevant data within the paper, Supporting Information files, or in an acceptable, public repository. Please add a citation to support this phrase or upload the data that corresponds with these findings to a stable repository (such as Figshare or Dryad) and provide and URLs, DOIs, or accession numbers that may be used to access these data. Or, if the data are not a core part of the research being presented in your study, we ask that you remove the phrase that refers to these data.

Reviewers' comments:

Reviewer's Responses to Questions

**Comments to the Author**

1. Is the manuscript technically sound, and do the data support the conclusions?

Reviewer #1: Partly

Reviewer #2: Partly

2. Has the statistical analysis been performed appropriately and rigorously? 

Reviewer #1: No

Reviewer #2: Yes

3. Have the authors made all data underlying the findings in their manuscript fully available?

Reviewer #1: Yes

Reviewer #2: Yes

4. Is the manuscript presented in an intelligible fashion and written in standard English?

Reviewer #1: No

Reviewer #2: Yes

5. Review Comments to the Author

Reviewer #1: Line 38: the presented percentage might exaggerate the result as the observed outcome is only 1. Hence it is better if the author avoid calculating proportion for very small numbers.

The investigators included only sociodemographic factors in to the final multivariate model. However the exposure and clinical risk factors are more important for HBV and HCV coinfection. so it is better if the analysis is repeated with missed more relevant factors for decision making and policy purposes.

finally I think the conclusion is biased as the analysis and presentation lack the most relevant clinical and behavioral factors

Reviewer #2: Comments on the manuscript “HIV, hepatitis B virus, and hepatitis C virus Co-infection patients in Addis Ababa Antiretroviral Treatment Program, 2012-2018: a retrospective cohort study” PONE-D-21-34939

Line 1. HIV, hepatitis B virus, and hepatitis C virus Co-infection patients in Addis Ababa : Add “among “ between Co-infection and patients

Line 1-2 : HIV, hepatitis B virus, and hepatitis C virus Co-infection patients in Addis Ababa Antiretroviral Treatment Program, 2012-2018: a retrospective cohort study: How this study become a retrospective study as far as the authors took data in a one point time ? I felt it is a retrospective cross sectional study !.

The authors also considered the study was done in Addis Ababa, rather it was done among selected hospitals and it is good to revise the title in that sense.

Line 76-77, Study design and setting: the study design is not described well !! it mention the word retrospective only which is not adequate.

Line 90- 92: Direct acting drugs that threaten HCV are very expensive and not easily accessible, even for those who can 92 afford to pay for it. : There are national initiative in line with this and better to underscore that and your study could support that initiative.

Line 96. Study participants without viral hepatitis test results were excluded from the analysis: About 125 HIV patients had no HBSAg and HCV tests ? how can we assure bias happened from these group of HIV patients on Co-infection of HIV/HBV and HIV /HCV co-infection rate? You need to state this as a in the limitation part in more description.

Similarly, the authors did not know what types of rapid tests for HBSAg and HCV tests were implemented ? any effect of accuracy / sensitivity/ specificity issues on proportions of co-infections among the three hospitals ? Changes of test kits between time and years??

The authors did not mentioned the three hospitals in Addis Ababa and how many HIV patients each hospitals could have ? how many participants were enrolled from each hospital ? It is better to describe these under methods part.

It seems that the authors described the magnitude of HBSAg and HCV in the study site is moderately high, HIV-HBV co-infection was 5.96% (95% CI: 4.56-7.74) , HIV-HCV co-infection was 1.72% (95% CI: 1.03-2.83), and triple co-infection of HIV-HBV-HCV was 1(0.11%) . In reality this magnitude is determined for 7 years’ time (between 2012-2018). I think the authors should consider this while they describe the co-infection rate. Rather If they anticipate for yearly prevalence one can see trend of co-infections as far as they have adequate number of participants per year !.

Line 199-200 : We found that the magnitude of HIV-HBV co-infection was 5.96% and HIV-HCV co200 infection was 1.72% in adults ≥ 15 years in Addis Ababa.: how can we generalize your findings for Addis Ababa? Be specific to HIV patients and specific hospitals

Line 275 References: References

6. PLOS authors have the option to publish the peer review history of their article (what does this mean?). If published, this will include your full peer review and any attached files.

Reviewer #1: No

Reviewer #2: No

---

## [Author Response · Author response to Decision Letter 0]

11 Mar 2022

Point-by-point response to reviewer’s

Dear Editor and Reviewers

Thank you very much for taking time in reviewing our manuscript and providing valuable comments. We took your suggestions and comments into consideration and revised our manuscript accordingly. The point-by-point response presents as below. 

I. Editor comment 

Comment 1. Please ensure that your manuscript meets PLOS ONE's style requirements, including those for file naming. 

Response 1: As suggested we now organized our manuscript as per the PLOS ONE’s style and requirement and the naming formats. 

Comment 2. Please provide additional details regarding participant consent. Specifically, if the need for consent was waived by the ethics committee, please include this information. Thank you.

Response 2: Thank you for this valuable suggestion, statement on consent wavier included in the manuscript, under method-Ethics section on page 8 line 169 to 173 

Comment 3. We note that the grant information you provided in the ‘Funding Information’ and ‘Financial Disclosure’ sections do not match. 

Response 3: We now removed statements regarding funding from the manuscript and no grant given. 

Comment 4. Thank you for stating the following in the Acknowledgments Section of your manuscript: 

We declare that there was no direct fund that the authors received. However, the data collection fee at the three hospitals was covered through AHRI by Federal HIV/AIDS Prevention and Control Office. The source of fund had no role in the design or conduct of the study; the collection, management, analysis, and interpretation of the data; or the preparation, review, or approval of the manuscript. The authors had full access to all the data in the study and takes the responsibility for the integrity of the data and the accuracy of the data analysis. The authors declares that there was no fund received from any organization. 

Response 4: We included amendment statements on the cover letter “The author(s) received no specific funding for this work” 

Comment 5. We note that you have indicated that data from this study are available upon request. PLOS only allows data to be available upon request if there are legal or ethical restrictions on sharing data publicly. For more information on unacceptable data access restrictions, please see http://journals.plos.org/plosone/s/data-availability#loc-unacceptable-data-access-restrictions. 

Response 5: We agree with data availability statement and amended on cover letter as

“All data are fully available without restriction”

Comment 6. We note that you have stated that you will provide repository information for your data at acceptance. Should your manuscript be accepted for publication, we will hold it until you provide the relevant accession numbers or DOIs necessary to access your data. If you wish to make changes to your Data Availability statement, please describe these changes in your cover letter and we will update your Data Availability statement to reflect the information you provide.

Response 6: Data Availability statement amended on the cover letter “Yes - all data are fully available without restriction “

Comment 7. We note that you have included the phrase “data not shown” in your manuscript. Unfortunately, this does not meet our data sharing requirements. PLOS does not permit references to inaccessible data. We require that authors provide all relevant data within the paper, Supporting Information files, or in an acceptable, public repository. Please add a citation to support this phrase or upload the data that corresponds with these findings to a stable repository (such as Figshare or Dryad) and provide and URLs, DOIs, or accession numbers that may be used to access these data. Or, if the data are not a core part of the research being presented in your study, we ask that you remove the phrase that refers to these data.

Response 7: The statement “data not shown” is removed from page 11 line 204 as it was also suggested by reviewer 1. 

II. Review Comments to the Author

Reviewer # 1

Comment 1: Line 38: the presented percentage might exaggerate the result as the observed outcome is only 1. Hence it is better if the author avoids calculating proportion for very small numbers.

Response 1: Thank you very much for the comment and suggestions. It is true, if a percentage is presented out of a very small number it gives wrong impression. However, in our case, we have a total of 873 people tested for both (HBV and HCV) and the triple coinfection found is only 1/873(0.11%). We still believe that showing the percent does not exaggerate. However, our finding of triple coinfection is very small and may not be a big concern and we omitted from the analysis on page 3 line 47, page 11 line 205. 

Comment 2: The investigators included only sociodemographic factors into the final multivariate model. However, the exposure and clinical risk factors are more important for HBV and HCV coinfection. so it is better if the analysis is repeated with missed more relevant factors for decision making and policy purposes.

Response 2. We highly appreciate the comment. We used secondary data; the analysis was limited to the available clinical and sociodemographic variables and we acknowledged this limitation on page 18 line 286 to 289. 

Regarding the inclusion of clinical factors such as WHO staging, CD4 cell count, hemoglobin etc these could be an effect of disease (due to HIV, viral hepatitis, or TB) rather than a risk factor. However, as suggested, we included WHO staging, CD4 in the model and found still sex as an independent predictor. Hence, we would like to retain the risk factor analysis as it is. 

Comment 3: I think the conclusion is biased as the analysis and presentation lack the most relevant clinical and behavioral factors.

Response 3. We understand the concern, however our conclusion emphasized only the overall magnitude of viral hepatitis coinfection and higher coinfection in men. We explained the limitation of the data and hence we believe our conclusion is reasonable. 

Reviewer #2: 

Comments 1: on the manuscript “HIV, hepatitis B virus, and hepatitis C virus Co-infection patients in Addis Ababa Antiretroviral Treatment Program, 2012-2018: a retrospective cohort study” PONE-D-21-34939 

Line 1. HIV, hepatitis B virus, and hepatitis C virus Co-infection patients in Addis Ababa : Add “among “ between Co-infection and patients

Response 1: Thank you for your suggestion. We have now added the world “among” on the title on page 1 line 1 and the title has been rephrased accordingly. 

Comment 2 : Line 1-2 : How this study become a retrospective cohort study as far as the authors took data in a one point time? I felt it is a retrospective cross-sectional study.

Response 2: Thank you for this comment. As we used data only at enrollment the correct labelling is retrospective cross-sectional study. We have corrected that in the revised manuscript as the changes are indicated in the title and method sections. 

Comment 3: The authors also considered the study was done in Addis Ababa, rather it was done among selected hospitals and it is good to revise the title in that sense.

Response 3: Thank you very much for the comment. We modified the title accordingly on page 1 line 1 

As you suggested, we indicated that the study was done in selected three hospitals in the method section to be more specific: on page 5 line 88-89 

Comment 4: Line 76-77, Study design and setting: the study design is not described well !! it mention the word retrospective only which is not adequate. 

Response 4: We agree with the comment and included the word “cross-sectional” on page 5 line 87 

Comment 5 : Line 90- 92: Direct acting drugs that threaten HCV are very expensive and not easily accessible, even for those who can afford to pay for it: There are national initiative in line with this and better to underscore that and your study could support that initiative. 

Response 5: We agree with the comment completely. There are national initiatives regarding viral hepatitis C screening and treatment. Now statement added on national initiatives on page 5 line 103-109. 

Comment 6: Line 96. Study participants without viral hepatitis test results were excluded from the analysis: About 125 HIV patients had no HBSAg and HCV tests? how can we assure bias happened from these group of HIV patients on Co-infection of HIV/HBV and HIV /HCV co-infection rate? You need to state this as a in the limitation part in more description.

Response 6: We agree with the comment. This was indicated in the limitation section on page 18 line 281 to 286 

Comment 7: Similarly, the authors did not know what types of rapid tests for HBSAg and HCV tests were implemented? any effect of accuracy /sensitivity/ specificity issues on proportions of co-infections among the three hospitals? Changes of test kits between time and years?? 

Response 7: The type of test used during the data collection and their sensitivity and specificity was not available. This remains the limitation of the study as indicated on page 18 line 289 to 291.

Comment 8 : The authors did not mentioned the three hospitals in Addis Ababa and how many HIV patients each hospitals could have? how many participants were enrolled from each hospital? It is better to describe these under methods part,

Response 8: Thank you for the valuable comment. As suggested, we indicated the name of the hospitals and their respective sample on page 6 line 124 to 125 

Comment 9. It seems that the authors described the magnitude of HBSAg and HCV in the study site is moderately high, HIV-HBV co-infection was 5.96% (95% CI: 4.56-7.74) , HIV-HCV co-infection was 1.72% (95% CI: 1.03-2.83), and triple co-infection of HIV-HBV-HCV was 1(0.11%) . In reality this magnitude is determined for 7 years’ time (between 2012-2018). I think the authors should consider this while they describe the co-infection rate. Rather If they anticipate for yearly prevalence one can see trend of co-infections as far as they have adequate number of participants per year !.

Response 9: As more than 80% of the sample was from year 2015-2018 the sample was not enough for trend analysis. The purpose of our study was not to see trend analysis rather to see the burden of viral hepatitis in ART clients among those who have got test access. 

Comment 10: Line 199-200: We found that the magnitude of HIV-HBV co-infection was 5.96% and HIV-HCV co200 infection was 1.72% in adults ≥ 15 years in Addis Ababa.: how can we generalize your findings for Addis Ababa? Be specific to HIV patients and specific hospitals. 

Response 10. Agree with the suggestion, we did not generalize for Addis Ababa, however, taking into consideration the homogeneity of ART users in Addis Ababa, and the likely hood of viral hepatitis transmission time (>90% at childhood) the finding can give a clue on the magnitude of viral hepatitis among ART users in Addis Ababa

Comment 11 : Line 275 References: 

Response 11 : References: this was corrected accordingly on page 19 line 304

---

## [Decision Letter · Decision Letter 1]

5 Apr 2022

HIV, Hepatitis B Virus, and Hepatitis C Virus Co-infection Among HIV Positives in Antiretroviral Treatment Program in Selected Hospitals in Addis Ababa: a Retrospective Cross-sectional Study

PONE-D-21-34939R1

Dear Eleni Seyoum,

We’re pleased to inform you that your manuscript has been judged scientifically suitable for publication and will be formally accepted for publication once it meets all outstanding technical requirements.

Kind regards,

Desalegn Admassu Ayana, Ph.D

Academic Editor

PLOS ONE

Additional Editor Comments (optional):

Reviewers' comments:

Reviewer's Responses to Questions

**Comments to the Author**

1. If the authors have adequately addressed your comments raised in a previous round of review and you feel that this manuscript is now acceptable for publication, you may indicate that here to bypass the “Comments to the Author” section, enter your conflict of interest statement in the “Confidential to Editor” section, and submit your "Accept" recommendation.

Reviewer #1: All comments have been addressed

2. Is the manuscript technically sound, and do the data support the conclusions?

Reviewer #1: Yes

3. Has the statistical analysis been performed appropriately and rigorously? 

Reviewer #1: Yes

4. Have the authors made all data underlying the findings in their manuscript fully available?

Reviewer #1: Yes

5. Is the manuscript presented in an intelligible fashion and written in standard English?

Reviewer #1: Yes

6. Review Comments to the Author

Reviewer #1: All comments are addressed

manuscript technically sound, and do the data support the conclusions

Limitations of this research is also indicated.

7. PLOS authors have the option to publish the peer review history of their article (what does this mean?). If published, this will include your full peer review and any attached files.

Reviewer #1: **Yes: **Regassa, LD.

---

## [Editor Report · Acceptance letter]

14 Apr 2022

PONE-D-21-34939R1 

HIV, Hepatitis B Virus, and Hepatitis C Virus Co-infection Among HIV Positives in Antiretroviral Treatment Program in Selected Hospitals in Addis Ababa: a Retrospective Cross-sectional Study 

Dear Dr. Seyoum:

I'm pleased to inform you that your manuscript has been deemed suitable for publication in PLOS ONE. Congratulations! Your manuscript is now with our production department. 

Kind regards, 

on behalf of

Dr. Desalegn Admassu Ayana 

Academic Editor

PLOS ONE